# High-Precision Log-Ratio Spot Position Detection Algorithm with a Quadrant Detector under Different SNR Environments

**DOI:** 10.3390/s22083092

**Published:** 2022-04-18

**Authors:** Li Huo, Zhiyong Wu, Jiabin Wu, Shijie Gao, Yunshan Chen, Yinuo Song, Shuaifei Wang

**Affiliations:** 1Changchun Institute of Optics, Fine Mechanics and Physics, Chinese Academy of Sciences, Changchun 130033, China; huoli18@mails.ucas.ac.cn (L.H.); wujb@ciomp.ac.cn (J.W.); gaoshijie@ciomp.ac.cn (S.G.); yiyunsn@163.com (Y.C.); songyinuo17@mails.ucas.ac.cn (Y.S.); wangshuaifei20@mails.ucas.ac.cn (S.W.); 2University of Chinese Academy of Sciences, Beijing 100049, China

**Keywords:** quadrant detector, log-ratio algorithm, gaussian spot

## Abstract

In atmospheric laser communication, a beam is transmitted through an atmospheric channel, and the photocurrent output from a quadrant detector (QD) used as the tracking sensor fluctuates significantly. To ensure uninterrupted communication and to adapt to such fluctuations, in this paper we apply logarithmic amplifiers to process the output signals of a QD. To further improve the measurement accuracy of the spot position, we firstly propose an integral infinite log-ratio algorithm (IILRA) and an integral infinity log-ratio algorithm based on the signal-to-noise ratio (BSNR-IILRA) through analysis of the factors influencing the measurement error considering the signal-to-noise ratio (SNR) parameter. Secondly, the measurement error of the two algorithms under different SNRs and their variations are analyzed. Finally, a spot position detection experiment platform is built to correctly and efficiently verify the two algorithms. The experimental results show that when the SNR is 54.10 dB, the maximum error and root mean square error of the spot position of the IILRA are 0.0054 mm and 0.0039 mm, respectively, which are less than half those of the center approximation algorithm (CAA). When the SNR is 23.88 dB, the maximum error and root mean square error of the spot position of the BSNR-IILRA are 0.0046 mm and 0.0034 mm, respectively, which are one-thirtieth and one-twentieth of the CAA, respectively. The spot position measurement accuracy of the two proposed algorithms is significantly improved compared with the CAA.

## 1. Introduction

A QD is a position sensor based on a photoelectric effect. Due to its advantages of being high resolution, fast response, and low cost [1,2,3], it has become a popular non-contact measurement instrument. Therefore, it is widely used in laser guidance [4], laser tracking [5], sub-nanometer measurements [6], space laser communication [7], and synchronous optical position detection [8], as well as in other fields.

General operational amplifiers and logarithmic operational amplifiers are often used to amplify the output current of the QD to the voltage signal for the facilitation of subsequent processing. The Δ/Σ algorithm using operational amplifiers is generally applied to position or angle measurement fields that require a high precision and high resolution, such as sub-nanometer measurements, autocollimators, and optical tweezers. The traditional Δ/Σ algorithm uses the center approximation method to position the centroid of the light spot [9]. However, there is a nonlinear relationship between the estimated position and the actual position of the center approximation method, which limits the measurement accuracy and range. To reduce the influence of non-linearity, a polynomial fitting method was proposed [10]. In [11], the integral infinite method and the Boltzmann method were proposed and a method of function fitting by combining the two methods was used to improve the measurement accuracy. Based on the opposite error characteristics of the center approximation method and the polynomial fitting method, the two methods were combined in [12] to effectively improve the measurement range of a QD. The Kalman filter method [13] and the cross-correlation method [14] were applied to improve the SNR, thereby improving the measurement accuracy of the spot position. Machine learning [15] and artificial neural networks [16] are also effective options. Although these algorithms have high accuracy and a high resolution, the method by which the signal is amplified limits the dynamic range.

Due to the advantages of a high bandwidth and a wide dynamic range [17], the log-ratio algorithm is generally used in fields such as atmospheric laser communication and synchronous optical position detection. In [18], a traditional log-ratio algorithm in a standard mode and a cross mode was proposed. In [8,19] the Δ/Σ algorithm and the log-ratio algorithm were compared and their linear range and sensitivity were analyzed. However, these algorithms did not derive a closed expression for the spot position, and the effect of noise on the log-ratio algorithm was not reported. Random noise produces random errors, which can be improved by filtering; the DC component of noise increases the output current of the QD, which produces systematic positioning errors. This limits the measurement accuracy of the log-ratio algorithm at a high dynamic range.

Therefore, for the purpose of solving the existing problems of the above log-ratio algorithm, in this paper we propose an IILRA and the approximate closed expression of the spot centroid position is derived. On this basis, the influencing factors of the positioning error are analyzed. The IILRA has a large positioning error at low SNRs. Therefore, the SNR is considered in a closed expression, and a BSNR-IILRA is proposed, which effectively improves the measurement accuracy at a low SNR. The IILRA and BSNR-IILRA achieve a higher measurement accuracy at high and low SNRs, respectively, which is confirmed by both simulation and experimental results.

In this paper, we present the working principle of QD in Section 2. In Section 3, the IILRA and BSNR-IILRA are discussed, and the positioning errors of both algorithms are analyzed. In Section 4, the spot centroid position measurement system and experiments are described, and the experimental verifications of the two algorithms are given. In Section 5, we summarize the conclusions.

## 2. The Principle of the Log-Ratio Algorithm of a QD

Figure 1 shows the principle of a QD. When a laser beam is irradiated on the QD, each quadrant of the QD generates a corresponding photocurrent according to the energy of the light spot in the respective area. When the laser spot moves on the QD, the spot energy in each quadrant changes, resulting in corresponding changes in the photocurrent of each quadrant. To improve the dynamic range, logarithmic operational amplifiers are used to amplify the photocurrent of the QD. The output voltage of each logarithmic amplifier is converted into a digital signal by an analog-to-digital converter. Finally, the approximate location of the spot centroid is calculated by a field programmable gate array (FPGA).

The traditional log-ratio algorithm takes the calculated value as the relative position of the spot centroid [18], and they can be expressed as:(1)$$\left\{ \begin{array}{l} \delta_{x}=\mathrm{lg}\frac{I_{A}+I_{D}}{I_{B}+I_{C}}\\ \delta_{y}=\mathrm{lg}\frac{I_{A}+I_{B}}{I_{C}+I_{D}} \end{array}\right.$$
(2)$$\left\{ \begin{array}{l} \delta_{x\text{-}cross}=\mathrm{lg}\frac{I_{A}}{I_{C}}\\ \delta_{y\text{-}cross}=\mathrm{lg}\frac{I_{B}}{I_{D}} \end{array}\right.$$
where *δ_x_*, *δ_x-cross_* and *δ_y_*, *δ_y-cross_* are the calculated values of the centroid position of the light spot in the x-direction and the y-direction, respectively, and *I**_A_*, *I**_B_*, *I**_C_*, and *I**_D_* are the photocurrents generated by each quadrant, respectively. Equation (1) was applied to the standard mode, as shown in Figure 2a, and Equation (2) was applied to the cross mode rotated by 45 degrees, as shown in Figure 2b. As shown in Figure 2c, there was a non-linear relationship between the calculated value and the actual position of both methods. Only when the centroid of the spot was close to the center of the measurement area did the calculated value have a linear relationship with the actual position. The non-linearity became apparent as the spot centroid moved toward the edge of the measurement region. In the cross mode, the spot centroid position was quickly calculated using the output voltage of the logarithmic amplifier. As the x-direction and the y-direction were not independent of each other in the cross mode, the calculated value *δx* was affected by *y*_0_. Therefore, the measurement results were relatively accurate only on the coordinate axis and had a significant deviation in the two-dimensional measurement.

## 3. Theoretical Method of the Spot Position Measurement

### 3.1. Integral Infinite Log-Ratio Algorithm

The laser beam is irradiated on a QD after passing through the optical system. The photocurrent of each quadrant could, therefore, be expressed as:(3)$$I_{i}=\eta \iint_{S_{i}}h(x,y)dxdy,\;\;\;\;\;\;i=A,B,C,D$$
where *h*(*x, y*) is the energy distribution of the spot when the centroid of the spot is located at (*x*_0_*, y*_0_), *η* is the response of the QD, and *S_i_* is the area of each quadrant. Assuming that the spot energy distribution obeyed a Gaussian distribution, the energy distribution of the light spot *h*(*x, y*) could be expressed as:(4)$$h(x,y)=\frac{2P}{\pi \omega^{2}}\exp\left[-2\frac{{\left(x-x_{0}\right)}^{2}+{\left(y-y_{0}\right)}^{2}}{\omega^{2}}\right]$$
where *ω* is the beam waist radius of the Gaussian spot, (*x*_0_*, y*_0_) is the centroid coordinate of the spot, and *P* is the total energy of the spot. As the x-direction and the y-direction were independent of each other and had the same principle, the following discussion takes the x-direction as an example to simplify the derivation process. Assuming that the responses of each quadrant of the QD were consistent, the calculated value of the spot centroid of the log-ratio algorithm was rewritten as:(5)$$\delta_{x}=\mathrm{lg}\frac{\int_{d/2}^{\sqrt{R^{2}-x^{2}}}\int_{d/2}^{\sqrt{R^{2}-d^{2}/4}}h(x,y)dxdy+\int_{-\sqrt{R^{2}-x^{2}}}^{-d/2}\int_{d/2}^{\sqrt{R^{2}-d^{2}/4}}h(x,y)dxdy}{\int_{d/2}^{\sqrt{R^{2}-x^{2}}}\int_{-\sqrt{R^{2}-d^{2}/4}}^{-d/2}h(x,y)dxdy+\int_{-\sqrt{R^{2}-x^{2}}}^{-d/2}\int_{-\sqrt{R^{2}-d^{2}/4}}^{-d/2}h(x,y)dxdy}$$
where *R* is the radius of the QD and *d* is the gap width. To obtain a simplified expression of the calculated value of the spot position, the integral infinite method was used. When *w* << *R*, the integral limit was changed to infinity and the effect of the gap was ignored; therefore, Equation (5) could be simplified as:(6)$$\delta_{x}\approx \mathrm{lg}\frac{\int_{-\infty }^{+\infty }\int_{0}^{+\infty }h(x,y)dxdy}{\int_{-\infty }^{+\infty }\int_{-\infty }^{0}h(x,y)dxdy}=\mathrm{lg}\frac{1+\mathrm{erf}(\sqrt{2}x_{0}/\omega )}{1-\mathrm{erf}(\sqrt{2}x_{0}/\omega )}$$
where erf(·) is the error function. To obtain the inverse function of Equation (6), the approximate position *x_a_* of the spot centroid for the IILRA could be written as:(7)$$x_{a}(\delta_{x})\approx \frac{\omega }{\sqrt{2}}{\mathrm{erf}}^{-1}\left(\frac{\exp(\delta_{x}\ln10)-1}{\exp(\delta_{x}\ln10)+1}\right)$$
where erf^−^^1^(·) is the inverse function of the error function erf(·). Figure 3a shows the relationship between the approximate position *x_a_* and the calculated value *δ_x_* for the IILRA. Figure 3b shows the error *e_IILRA_* caused by the integral infinity method.

### 3.2. Integral Infinite Logarithmic Ratio Algorithm Based on SNR

Noise can degrade the measurement accuracy of QDs [20]. Due to the characteristics of the logarithm, the measurement accuracy of the log-ratio algorithm is significantly affected by noise. The main sources of noise are background light and the dark current of the QDs. In actual measurement systems, random noise will generate random errors, but these can be improved by filtering. The DC component of noise increases the output photocurrents in each quadrant, creating systematic errors that affected the measurement accuracy. Therefore, it was necessary to compensate for the DC component of the noise by an algorithm. The calculated value *δ_x_* was rewritten as:(8)$$\delta_{x}=\mathrm{lg}\frac{I_{SA}+I_{NDCA}+I_{SD}+I_{NDCD}}{I_{SB}+I_{NDCB}+I_{SC}+I_{NDCC}}$$
where *I_SA_*, *I_SB_*, *I_SC_*, and *I_SD_* are the photocurrents generated by the signal light in each quadrant, and *I_NDCA_*, *I_NDCB_*, *I_NDCC_*, and *I_NDCD_* are the DC components of the noise in each quadrant. The main sources of noise were from the dark current, background light, and back-end circuits. From Equation (10), it could be seen that, due to the influence of noise, the numerator and denominator increased by the same value, resulting in a decrease in |*δ*_x_|, which affected the measurement accuracy. Generally speaking, as the noise source of each quadrant of the QD was independent of one another and had a good consistency, the background light could be considered to be uniformly distributed, so it could be expressed as:(9)$$I_{NDCA}\approx I_{NDCB}\approx I_{NDCC}\approx I_{NDCD}\approx I_{NDC}$$

After filtering, and ignoring the influence of random noise, the SNR could be approximated as:(10)$$SNR\approx I_{S}^{2}/4I_{NDC}^{2}$$
where *I_S_* = *I_SA_* + *I_SB_ + I_SC_ + I_SD_* is the total photocurrent generated by the beacon light incident on the QD, $I_{S}^{2}$ is the total signal power, and 4$I_{NDC}^{2}$ is the total noise power. By substituting Equations (3), (4), (9) and (10) into Equation (8), and by using the integral infinity method, the calculated value could be approximated as:(11)$$\delta_{x}\approx \mathrm{lg}\frac{1+\mathrm{erf}(\sqrt{2}x_{0}/\omega )\frac{1}{1+2/\sqrt{SNR}}}{1-\mathrm{erf}(\sqrt{2}x_{0}/\omega )\frac{1}{1+2/\sqrt{SNR}}}$$

Assuming that the QD radius was 1.5 mm, the gap width was 0.02 mm, and the beam waist radius ω was 0.55 mm, the measurement range moved from (−0.5 mm, 0) to (+0.5 mm, 0) along the x-direction. As shown in Figure 4, in the process of gradually increasing the SNR, the shape of the curve gradually changed from an “S” shape to an inverted “S” shape. The non-linearity increases as the SNR approached 0 or infinity. Therefore, when the SNR was moderate, the linearity of the calculated value was high, and it was appropriate to use the center approximation method to locate the spot position. In other cases, geometric approximation methods were a better choice.

By introducing the SNR parameter, the approximate position *x_a_* of the spot centroid of the BSNR-IILRA could be expressed as:(12)$$x_{a}(\delta_{x},SNR)\approx \frac{\omega }{\sqrt{2}}{\mathrm{erf}}^{-1}\left((1+2/\sqrt{SNR})\frac{\exp(\delta_{x}\ln10)-1}{\exp(\delta_{x}\ln10)+1}\right)$$

When the SNR tended toward positive infinity—that is, the influence of noise was not considered—Equation (12) was equivalent to Equation (7), which verified the correctness of Equation (12). Figure 5a shows the relationship between the approximate position *x_a_* and the calculated value *δ_x_* for the BSNR-IILRA. Figure 5b shows the error *e**_BSNR-IILRA_* caused by the integral infinity method.

The positioning error *e_x_* was the difference between the approximate position *x_a_* and the real position *x*_0_ of the spot centroid:(13)$$e_{x}=x_{a}-x_{0}$$

According to the mathematical model of the IILRA, the positioning error was mainly affected by two factors. One was ignoring the influence of noise and the other was using the integral infinity method to approximate the centroid of the spot. Due to the influence of the DC component of noise, |*δ_x_*| became smaller. Using the IILRA to calculate the centroid position of the light spot resulted in a smaller |*x_a_*|, and the lower the SNR, the larger the positioning error. Using the integral infinity method caused the photocurrent ratio of each quadrant to be larger than the measured value, which increased |*x_a_*| within the detection range, as shown in Figure 3b and Figure 5b. The higher the SNR, the larger the positioning error. As shown in Figure 6a, the positioning error of the IILRA was larger at a low SNR, mainly due to the DC component of noise. At a high SNR, it was mainly due to the use of the integral infinity method. The positioning errors caused by the two factors were in opposite directions and several of the positioning errors caused by the two factors canceled each other out. Therefore, the positioning error of the IILRA first decreased and then increased with an increase in the SNR. As shown in Figure 6b, as the BSNR-IILRA compensated for the positioning error caused by the DC component of noise, the error was mainly due to the use of the integral infinity method, which was not significantly affected by the SNR. However, at a high SNR, the positioning errors caused by the two factors did not cancel each other out and the positioning error of the BSNR-IILRA was slightly larger than that of the IILRA.

## 4. Experimental System and Results

To verify the effectiveness and correctness of the two proposed algorithms, an experimental system was designed, as shown in Figure 7. A 1550 nm laser beam was propagated through a collimator and a focusing lens, and then focused onto a QD (EOSPACE FM-IGA-030-QD, 1.5 mm detector radius, 0.02 mm gap). By adjusting the one-dimensional nanoscale displacement platform (displacement platform: PI N-664.3A; controller: PI E-861.1A1), the position of the spot centroid on the QD changed. Four logarithmic amplifiers were used to amplify the output current of the QD. The output voltages of these amplifiers were then collected by a digital-to-analog converter. Finally, the centroid position of the spot was calculated according to a FPGA (Cyclone 4 EP4CE30) computing module. The SNR was adjusted by changing the power of the laser (Teraxion, PS-NLL-E-1549.32-80-03, 1550 nm). A block diagram of the spot position measurement system is shown in Figure 8.

First, the distance between the convex lens and the QD was changed, and the spot radius was adjusted to 0.55 mm. The photocurrent generated by the QDs in the presence of no signal light was measured as the DC component of noise. The theoretical position of the spot centroid was obtained by a one-dimensional nano-displacement platform. The spot centroid was then moved from (−0.5 mm, 0) to (+0.5 mm, 0) along the x-direction in steps of 0.02 mm, and the data were collected for each position 100 times. The SNR was changed by adjusting the laser power and the data at different SNRs were collected. *δ_x_* was calculated from the collected data. The IILRA and BSNR-IILRA were then separately used to obtain the approximate positions of the spot centroid and the corresponding positioning errors. Figure 9 shows the experimental results of the positioning errors of the two algorithms. When comparing Figure 6 with Figure 9, it can be seen that the experimental results at the center of the measurement area agreed well with the simulation results. At the edge of the measurement area, the experimental curve had the same trend as the simulation curve, but with a few deviations. This was because the spot distribution was not an ideal Gaussian distribution, and the logarithmic amplifier had a greater sensitivity to current changes when the input current was small.

To more accurately evaluate the improvement in the algorithm on the detection accuracy of the spot position, the maximum error *e_xMAX_* and the root mean square error *e_xRMSE_* of the positioning error were introduced to evaluate the maximum and average degree of the approximate position of the spot centroid deviating from the actual position within the measurement range, respectively. The *e_xMAX_* and *e_xRMSE_* are defined as:(14)$$e_{xMAX}=\max(\left|e_{xi}\right|)$$
(15)$$e_{xRMSE}=\sqrt{\sum_{i=1}^{N}e_{xi}^{2}/N}$$
where *e_xi_* is the positioning error of the different positions in the detection area. Figure 10 shows the *e_xMAX_* and *e_xRMSE_* of the two algorithms under different SNRs, where the experimental results agreed well with the simulation. The *e_xMAX_* and *e_xRMSE_* of the IILRA first decreased and then increased with an increase in the SNR, which was consistent with the previous analysis. The *e_xMAX_* and *e_xRMSE_* took their minimum values when the SNR was approximately 48.60 dB and 47.64 dB, respectively. The BSNR-IILRA effectively reduced the *e_xMAX_* and *e_xRMSE_* at a low SNR, but slightly increased the *e_xMAX_* and *e_xRMSE_* at a high SNR because the positioning errors caused by the two factors did not cancel each other out. When the SNR was 45.62 dB, the *e_xMAX_* of the two algorithms was equal. When the SNR was 41.60 dB, the *e_xRMSE_* of the two algorithms was equal. The *e_xMAX_* and *e_xRMSE_* of the BSNR-IILRA were smaller when the SNR was lower. The *e_xMAX_* and *e_xRMSE_* of the IILRA were smaller when the SNR was higher.

The positioning errors of the IILRA, BSNR-IILRA, and CAA were compared, as shown in Table 1. The *e_xMAX_* and *e_xRMSE_* of the IILRA both decreased at first and then increased with an increase in the SNR. When the SNRs are 54.10 dB and 64.07 dB, the *e_xMAX_* numbers of the IILRA were 0.0054 mm and 0.0070 mm, respectively; the *e_xRMSE_* numbers of the IILRA were 0.0039 mm and 0.0054 mm, respectively, which were lower than the BSNR-IILRA. When the SNR were 23.88 dB and 33.92 dB, the *e_xMAX_* numbers of the BSNR-IILRA were 0.0046 mm and 0.0064 mm, respectively, which were reduced by 96.36% and 87.48%, respectively, compared with the IILRA. The *e_xRMSE_* numbers of the BSNR-IILRA were 0.0034 mm and 0.0049 mm, which were reduced by 94.09% and 75.74%, respectively, compared with the IILRA. When the SNR was 44.00 dB, the IILRA had a smaller *e_xRMSE_*, whereas the BSNR-IILRA had a smaller *e_xMAX_*, which was consistent with the previous analysis. The approximate spot position of the CAA could be obtained as:(16)$$x_{a}=k\delta_{x}$$
where k is the scaling factor of the CAA. It can be seen from Table 1 that the IILRA and the BSNR-IILRA were superior to the CAA. The *e_xMAX_* and *e_xRMSE_* of the IILRA were half that of the CAA at 54.10 dB and 64.07 dB, respectively; the *e_xMAX_* and *e_xRMSE_* of the BSNR-IILRA were one-thirtieth and one-twentieth of that of the CAA at 23.88 dB.

## 5. Conclusions

In summary, the IILRA and BSNR-IILRA closed expressions were derived for the purpose of improving the measurement accuracy of spot position detection using the log-ratio algorithm with a QD. The factors influencing the positioning error were analyzed and the influence of the SNR on the two algorithms was discussed. The performances of the IILRA, BSNR-IILRA, and CAA were compared. The results of the simulation and the experiment showed that the positioning error of the IILRA was significantly affected by the SNR at a low SNR, but its positioning error was lower than that of the BSNR-IILRA at a high SNR. The BSNR-IILRA significantly reduced the positioning error at a low SNR and was less affected by the SNR; therefore, it ensured a certain measurement accuracy when the SNR changed. Due to the different characteristics of the two algorithms, the appropriate algorithm could be selected according to the working environment, or the two algorithms could be combined, and the algorithms could be adaptively selected according to the measured SNR to obtain a higher measurement accuracy in the entire dynamic range. As the application of logarithmic amplifiers to process signals provides a higher dynamic range, the two proposed algorithms have greater applicability to fields such as atmospheric laser communication and synchronized optical position detection.

## Figures and Tables

**Figure 1 sensors-22-03092-f001:**
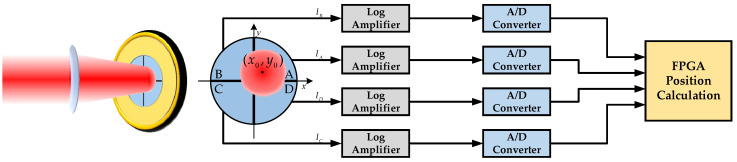
The working principle of the log-ratio algorithm.

**Figure 2 sensors-22-03092-f002:**
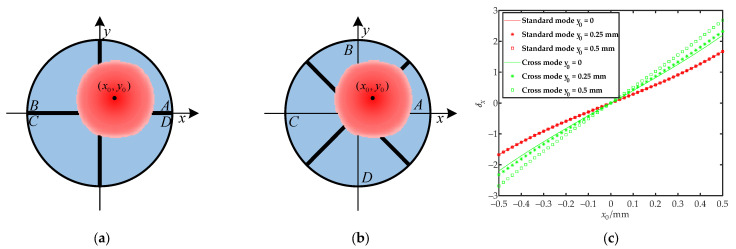
Modes for quadrant detector placement: (**a**) standard mode; (**b**) cross mode; (**c**) the relationship between the calculated value *δ_x_* and the theoretical value of spot position *x*_0_.

**Figure 3 sensors-22-03092-f003:**
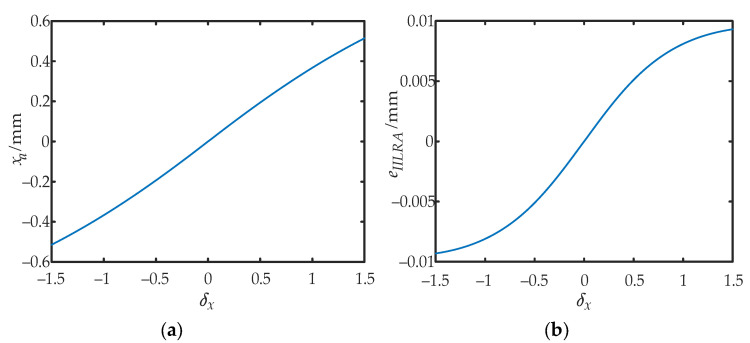
Approximate position *x_a_* of the IILRA and its error: (**a**) the relationship between the approximate position *x_a_* and the calculated value *δ_x_*; (**b**) error in the approximate position caused by the integral infinity method.

**Figure 4 sensors-22-03092-f004:**
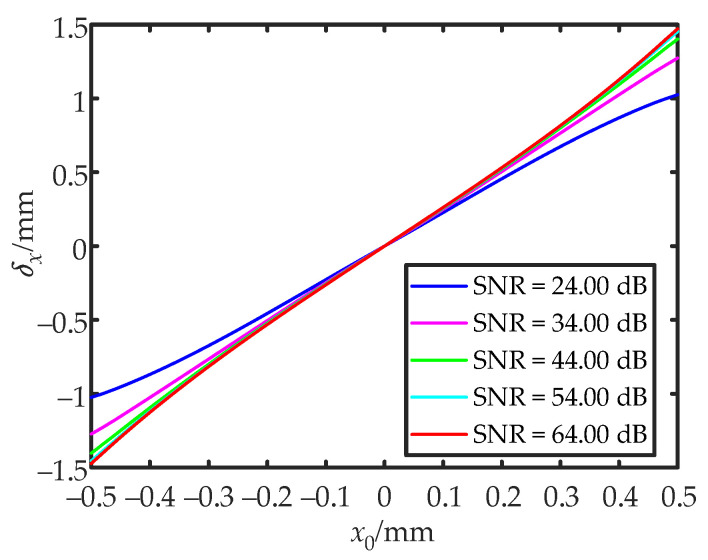
The relationship between the calculated value *δ_x_* and the theoretical value of the spot position *x*_0_.

**Figure 5 sensors-22-03092-f005:**
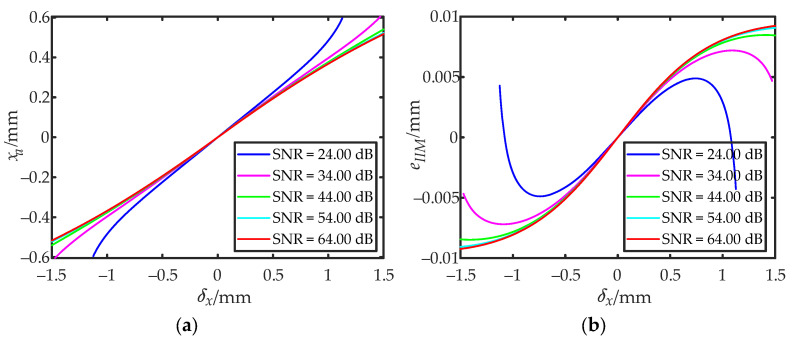
Approximate position *x_a_* of the BSNR-IILRA and its errors: (**a**) the relationship between the approximate position *x_a_* and the calculated value *δx* at different SNRs; (**b**) error in the approximate position caused by the integral infinity method at different SNRs.

**Figure 6 sensors-22-03092-f006:**
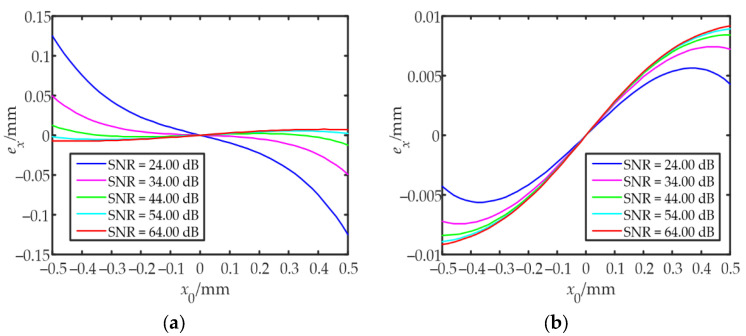
Simulation results of the positioning errors of the two algorithms: (**a**) the positioning errors of the IILRA; (**b**) the positioning errors of the BSNR-IILRA.

**Figure 7 sensors-22-03092-f007:**
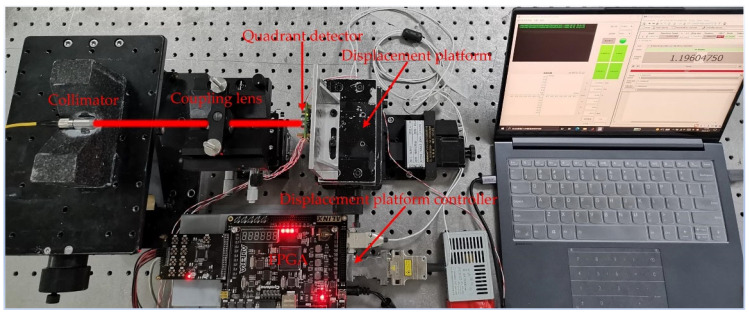
Measurement system of a spot position based on a QD.

**Figure 8 sensors-22-03092-f008:**
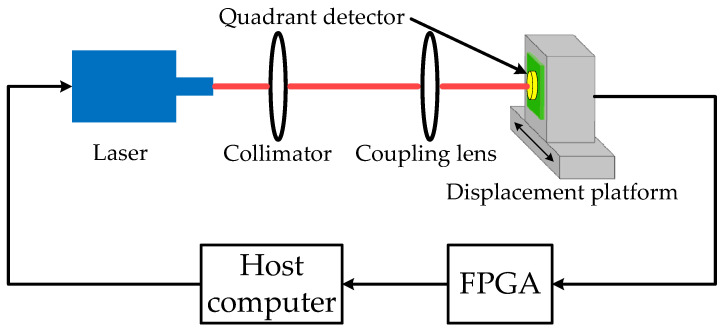
Block diagram of the spot position measurement system.

**Figure 9 sensors-22-03092-f009:**
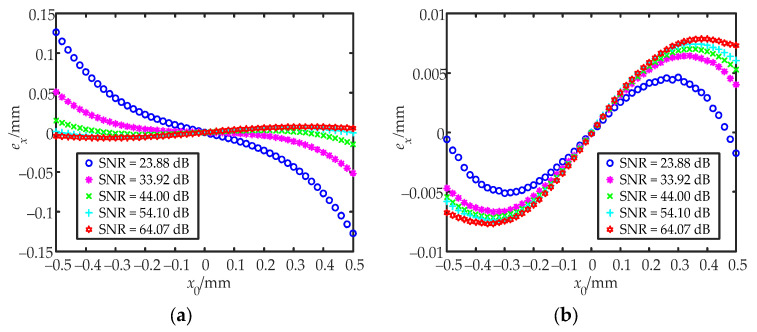
Experimental results of the positioning errors of the two algorithms: (**a**) the positioning errors of the IILRA; (**b**) the positioning errors of the BSNR-IILRA.

**Figure 10 sensors-22-03092-f010:**
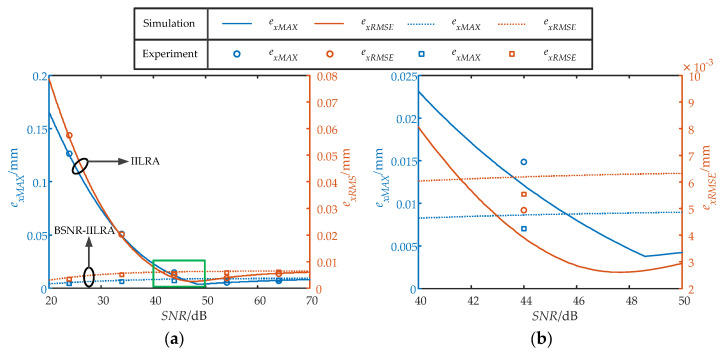
The maximum error *e_xMAX_* and the root mean square error *e_xRMSE_* of the two algorithms under different SNRs: (**a**) SNR range from 20 dB to 70 dB; (**b**) SNR range from 40 dB to 50 dB.

**Table 1 sensors-22-03092-t001:** Comparisons of the *e_xMAX_* and *e_xRMSE_* of the three algorithms under different SNRs.

*SNR/*dB	*e_xMAX_/*mm			*e_xRMSE_/*mm		
	IILRA	BSNR-IILRA	CAA	IILRA	BSNR-IILRA	CAA
23.88	0.1265	0.0046	0.1399	0.0575	0.0034	0.0705
33.92	0.0511	0.0064	0.0519	0.0202	0.0049	0.0292
44.00	0.0149	0.0070	0.0168	0.0049	0.0055	0.0125
54.10	0.0054	0.0075	0.0132	0.0039	0.0059	0.0092
64.07	0.0070	0.0079	0.0189	0.0054	0.0062	0.0094

## Data Availability

Data supporting reported results can be obtained from the corresponding author.

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
