# Peer review of "High-Precision Log-Ratio Spot Position Detection Algorithm with a Quadrant Detector under Different SNR Environments"

_sensors, 2022, doi:10.3390/s22083092_

Round 1

Reviewer 1 Report

The manuscript by Li Huo et al. proposed a modified method for spot position detection with quadrant detectors. This method seems to extend the measurement capability to low SNR environments with high precision and the methodology is technically sound. I would see the study fit for publication after the authors address the following comments. 

1. There are typos in the manuscript, for example, "appliedto" in line 94-95, "SBR"?? in line 174,176, "Error! Reference ...." in line 198, and several subscripts for variables are not written properly. The authors should recheck them.
2. The proposed method is based on the information of SNR. How can accurately acquire the value of SNR? The authors use SNR = 23.98,33,98,... for simulation and measurement. How can the authors determine the SNR is 23.98 and not 24 or 24.5...? What is the STD for SNR and its sensitivity to the measurement results? 
3.  It is difficult to understand the measurement results have higher precision in low SNR than those in high SNR while in the BSNR-IILRA method (see Fig. 10). Is that means low SNR is preferred? Or there is a inflection point for the BSNR-IILRS like that for IILRA at SNR~47dB? 
4. About the significant digits of the results in table 1. Is the measurement uncertainty supports such number of significant digits?

Reviewer 2 Report

I have the following suggestions to improve the quality of the paper. 

  • The paper is written in an adverse English language with a lot of grammar and errors in sentence structures. This fact makes it difficult to fully understand the message which author wants to give in the paper. I suggest availing of the English editing service from the MDPI publisher. Without it, I would not be willing to accept the paper. 
  • The paper is not written in purely scientific language. In most places, ordinary language has been used. For instance, instead of writing like this, “Reference [11] proposed a method of function fitting and combined the integration infinite method and the Boltzmann method to improve the measurement accuracy.” Use this “In [11], a method of function fitting by combining integration infinite method and the Boltzmann method has been used to improve the measurement accuracy.” Also, provide references to both the methods.
  • I suggest removing the word Reference from the text and writing it in this way, “In [ref]. ”
  • If SNR has been defined in the Introduction section, then don’t use its full form in heading 3.2.
  • Line 168, an error message appears, “Error! Reference source not found.” Rectify this.
  • A few more recent studies related to the QD position sensors should be studied and added in the paper. Moreover, the device performance should be compared with the one obtained in this paper.

Round 2

Reviewer 2 Report

I am willing to accept the paper in its current form.